# Lean Body Mass, Muscle Architecture and Powerlifting Performance during Preseason and in Competition

**DOI:** 10.3390/jfmk9020089

**Published:** 2024-05-22

**Authors:** Konstantinos Tromaras, Nikolaos Zaras, Angeliki-Nikoletta Stasinaki, Thomas Mpampoulis, Gerasimos Terzis

**Affiliations:** 1Sports Performance Laboratory, School of Physical Education and Sport Science, National and Kapodistrian University of Athens, 11527 Athens, Greece; k.tromaras@phed.uoa.gr (K.T.); agstasin@phed.uoa.gr (A.-N.S.); thompamp@phed.uoa.gr (T.M.); 2Department of Physical Education and Sports Sciences, Democritus University of Thrace, 69100 Komotini, Greece; nzaras@phyed.duth.gr; 3Human Performance Laboratory, Department of Life Sciences, School of Life and Health Sciences, University of Nicosia, 2417 Nicosia, Cyprus

**Keywords:** body composition, ultrasonography, resistance training, muscle strength, muscle hypertrophy

## Abstract

Lean body mass (LBM) is correlated with powerlifting performance in athletes competing in different bodyweight classes. However, it remains unknown whether changes in LBM are correlated with performance changes in powerlifters preparing for a competition. The aim of this study was to investigate the changes in LBM and performance in powerlifters preparing for a competition. Eight male powerlifters (age 31.7 ± 9.8 years, height 1.77 ± 0.06 m, weight 99.2 ± 14.6 kg) and three female powerlifters (age 32.7 ± 16.3 years, height 1.54 ± 0.06 m, weight 66.6 ± 20.9 kg) participated in the study. The athletes followed individualized periodized training programs for 12 weeks, aiming to maximize their performance for the national championship. The maximum strength (1-RM) in the squat, bench press, and deadlift, body composition, handgrip strength, anaerobic power, quadriceps’ cross-sectional area and vastus lateralis muscle architecture were measured before and after the training period. Significant increases were found after the training period in the squat (5.8 ± 7.0%, *p* < 0.05), bench press (4.9 ± 9.8%, *p* = 0.05) and deadlift (8.3 ± 16.7%, *p* < 0.05). Significant correlations were found between the 1-RM and LBM before and after the training period (r > 0.75, *p* < 0.05). The changes in the 1-RM after the training intervention correlated with the changes in the total LBM (*p* < 0.05). These results suggest that individual changes in LBM due to systematic resistance training for a competition may dictate increases in the 1-RM strength in powerlifters.

## 1. Introduction

Powerlifting is a dynamic strength sport in which three multi-joint lifts are performed in competition: the back squat (SQ), the bench press (BP), and the deadlift (DL). Each lift represents approximately 36%, 24%, and 40%, respectively, of the total lifting performance [1]. Accepting specific judging criteria, the aim of each athlete is to lift the heaviest possible load during competition. The International Powerlifting League (IPL) organizes the raw, the classic raw and the equipped competitions, with the raw allowing the wearing of specific knee sleeves, belt, and wrist wraps, besides the singlet [2]. The present study is about the raw powerlifting competition.

The maximum muscle strength is the main parameter determining performance in powerlifting. Muscle strength is mainly determined by the muscle volume, besides other biomechanical, muscle architectural and neural factors [3]. Powerlifters compete in specific bodyweight categories; therefore, it is anticipated that heavier athletes would have greater muscle mass and would be generally stronger compared to lighter athletes. Indeed, strong correlations have been reported between powerlifting performance and different measures of muscle mass. For example, muscle thickness at several body sites measured with ultrasonography was highly correlated with SQ, BP, and DL performance in trained powerlifters [4]. Similar results were reported in 20 trained powerlifters when using ultrasonography to estimate the fat-free mass and muscle mass [5]. A recent study employed lean body mass (LBM) measurements via dual X-ray absorptiometry (DXA) as a surrogate of muscle mass and reported very high correlations between SQ, BP, and DL performance and the total LBM, i.e., r = 0.94, r = 0.88, r = 0.86, (*p* < 0.001), respectively, in trained powerlifters [6]. The results of this previous study were confirmed by a recent study by the same research group, also demonstrating high correlations between the LBM and performance in powerlifters [7].

The strong relationship between maximum strength and muscle mass in powerlifters suggests that training-induced increases in muscle mass may result in maximum strength increases, perhaps even in competition. However, to the best of our knowledge, training-induced changes in performance and LBM in powerlifters preparing for a competition have not yet been investigated. A recent study monitored a small number of powerlifters for a year, providing body composition measurements; however, the possible link between the training-induced changes in performance and LBM was not reported [7]. Therefore, it remains uncertain whether training-induced changes in LBM may predict performance changes in these athletes. This information would be of value to powerlifting athletes and coaches because they could aim for specific changes in LBM, which would induce specific strength increases in competition.

The skeletal muscle architecture (muscle thickness, pennation angle and fascicle length), as measured by ultrasonography, has been shown to effect athletic performance [8]. As described above, muscle thickness at specific body sites was strongly correlated with muscle strength in powerlifters [4,5]. Yet, some early evidence revealed that the vastus lateralis (VL) pennation angle was not correlated with powerlifting performance, while the fascicle length was only moderately correlated with powerlifting performance [4]. Previous studies reported increases in the muscle thickness, pennation angle and fascicle length in well-trained athletes [9,10] in response to heavy resistance and power training. However, the effect of specific powerlifting training on the muscle architecture and the possible link between these adaptations and powerlifting performance has not been investigated.

The aim of this study was to investigate the relationship between the training-induced changes in LBM and muscle architecture and the changes in performance in well-trained powerlifters preparing for a competition. It was hypothesized that the changes in LBM and muscle thickness would correlate with the changes in powerlifting performance.

## 2. Materials and Methods

### 2.1. Participants

Eleven experienced powerlifters, eight males (age 31.7 ± 9.8 years, height 1.77 ± 0.06 m, body mass 99.2 ± 14.6 kg, best total powerlifting performance in SQ 250 ± 59.6 kg, BP 155.6 ± 35.4 kg and DL 266.6 ± 41.1 kg) and three females (age 32.7 ± 16.3 years, height 1.54 ± 0.06 m, body mass 66.6 ± 20.9 kg, best total powerlifting performance in SQ 132.5 ± 26.3 kg, BP 68.3 ± 21 kg and DL 135.8 ± 12.3 kg) participated in this study. All the athletes had at least 3 years of competitive experience. Four athletes participated in at least two international powerlifting events each year. The athletes were healthy, with no musculoskeletal injuries, and they all received > 2 gr of protein per kilogram of bodyweight daily via a normal diet and food supplements. They were informed orally and in written form about the research procedures and the possible risks, and they provided written consent regarding their participation in the study. All the procedures were performed in accordance with the principles outlined in the 1975 Declaration of Helsinki, as revised in 2000. All the procedures were approved by the Bioethics Committee of the School of Physical Education and Sports Science of the National and Kapodistrian University of Athens (protocol number 1321/22-09-2021).

### 2.2. Study Design

Experienced powerlifters were trained for 12 weeks following their individualized periodized programs as they prepared for the national competition. Before (T1) and after (T2) the training period, the body composition, quadriceps’ cross-sectional area and VL muscle architecture, as well as the Wingate anaerobic bicycle test, the countermovement jump test (CMJ) and the handgrip test, were evaluated. The maximum strength (1-RM) in the SQ, BP and DL was measured at T1 during a training session. Specifically, the athletes visited the laboratory on two different days at T1. During the first day, the body composition analysis, the quadriceps’ cross-sectional area (CSA) and the VL muscle architecture were evaluated. The second day included the performance measurements. The final 1-RM measurements (T2) were obtained from the official records of the national competition. The laboratory measurements at T2 were obtained at 72 h after the national competition. The differences between the T1 and T2 measurements were statistically compared, while correlation analysis was used between the training-induced changes for all the variables.

### 2.3. Procedures

#### 2.3.1. Training

All the athletes completed 12 weeks of individualized periodized training programs designed by their coaches, aiming to maximize their powerlifting performance at the national competition. A careful review of the individual training logs revealed that all the training programs included three mesocycles. The training during the first 6 weeks aimed to enhance muscle hypertrophy and strength (4–5 sets, 3–8 reps, 80–93% of 1-RM). The second 4-week mesocycle aimed to increase maximum strength production in the three competitive lifts (4–5 sets, 1–2 reps, 95–100% of 1-RM). During the first and second mesocycles, the athletes performed each main lift once per week, 48 h apart, accompanied by 1–2 accessory exercises for the same muscle group. Finally, during the third 2-week tapering mesocycle, the training aimed to reduce fatigue and increase performance before the national competition.

#### 2.3.2. 1-RM Strength

Powerlifting performance was measured before the initiation of the training period (T1) at the training facility where each athlete was training, 3 days before the initiation of the 12-week training period, under the supervision of at least 2 of the researchers, according to the International Powerlifting League regulations [2]. All the athletes followed an individual warm-up consisting of static and dynamic stretching exercises and several repetitions with an unloaded barbell in each lift. Subsequently, the athletes performed 2–3 sets of 4–6 repetitions with incremental submaximal efforts and then single repetitions until they could lift the heaviest load. Three to five minutes of rest was allowed between attempts. This protocol was followed for all 3 lifts with 30 min of rest between lifts (first the SQ, then the BP and finally the DL). The highest load for each lift was used for the statistical analysis. Powerlifting performance at T2 was collected from the national championship records following the regulations of the IPL. The best of the three maximum attempts for each lift was used for the statistical analysis. The sum of all three lifts was used for the total lift performance of the athletes.

#### 2.3.3. Body Composition

Body composition was assessed 1 day before the initial 1-RM measurements (T1) and 2 days following the powerlifting competition (T2). The athletes were instructed to fast for 12 h and refrain from any strenuous exercise for 24 h prior to the measurements [11]. Body mass was evaluated on a body scale (Tanita BC-545n, Tokyo, Japan), and body height was measured with a stadiometer (Seca 213, Surrey, UK). After the evaluation of the anthropometric characteristics, body composition was assessed via dual X-ray absorptiometry (DXA; Prodigy Pro, General Electric, Madison, WI, USA). The Lunar enCORE v.18 software was used to determine the bone mineral density (BMD), bone mineral content (BMC), body fat mass, percentage body fat and percent total and regional LBM, and visceral fat. The intra-class correlation coefficient (ICC) for the body fat mass was 0.99 and for the total, legs, arms, and trunk LBM, it was 0.99, 0.99, 0.98, 0.98, and 0.98, respectively.

#### 2.3.4. Ultrasonography

Ultrasonography was performed immediately after the DXA scans, at both T1 and T2, on the dominant lower extremity, always by the same researcher. B-mode ultrasound (Logiq P9, General Electric, USA) images were obtained with a 10–12 MHz linear-array probe for the quadriceps’ femoris CSA, while for the VL, images were obtained with a 15 MHz linear-array probe. For the quadriceps’ CSA imaging, a line was marked from the center of the patella to the medial aspect of the anterior superior iliac spine and then an axial perpendicular line was drawn at 40% of this distance (proximal to the knee). The probe was moved transversely across the thigh, on this marked line, taking a continuous single view that pictured the entire CSA of the quadriceps [12]. The CSAs of the four quadriceps’ heads were analyzed every single time by the same researcher using an image analysis software (ImageJ v.1.53, U.S. National Institutes of Health, Bethesda, MA, USA). Analysis of the architecture of the VL was performed with the transducer placed longitudinally on the femur, oriented in parallel to the muscle fascicles and perpendicular to the skin. A line was drawn on the front and back of 40% of the distance from the center of the patella to the medial aspect of the anterior superior iliac spine to identify and capture the largest, continuous fascicle visualization. A continuous single view was taken by moving the probe along the marked, dashed line. The images were always analyzed by the same researcher for the muscle thickness, pennation angle, and fascicle length with image analysis software (ImageJ v.1.53, U.S. National Institutes of Health, Bethesda, MA, USA). The muscle thickness was defined as the distance between the superficial and deep aponeurosis, the fascicle angle as the angle of insertion of the muscle fascicles onto the deep aponeurosis, and the fascicle length as the fascicular path between the insertions of the fascicle onto the upper and deeper aponeurosis. The ICCs for the CSA of VL, RF, VI, VM and total quadriceps’ CSA were 0.96, 0.94, 0.95, 0.87 and 0.97, respectively. In terms of the repeatability of the entire procedure, including the locations of the imaging sites and calculation of the architectural parameters, a test–retest was performed on 10 participants on 2 separate days when the skin markings were completely removed. The ICCs with 95% CIs (2-way random effects with absolute agreement) were calculated [13]. The ICCs for the VL muscle thickness, fascicle angle and fascicle length were 0.97, 0.88 and 0.84, respectively.

#### 2.3.5. Handgrip Strength

On a different day, the athletes visited the laboratory for the performance measurements. After a warm-up on a stationary bike and dynamic stretching, the handgrip strength was evaluated first, using a hydraulic Jamar hand dynamometer (Jamar, Patterson Medical, Warrenville, IL, USA). The athletes were in a standing position and the elbow was in full extension [14]. The handgrip strength was measured in both upper extremities, allowing 3 efforts for each hand with a 1 min rest between attempts. The highest strength value for the sum of the best performance of both hands was used in the statistical analysis. The ICC for the handgrip strength was calculated earlier in our laboratory (ICC = 0.96).

#### 2.3.6. Countermovement Jump

After the handgrip strength measurements, the athletes performed two CMJs with submaximal intensity. Subsequently, they performed two maximal CMJs, with 2 min of rest between each jump, on a force platform (Applied Measurements Ltd. Co., Reading, UK; WP1000, 1 kHz sampling frequency) with arms akimbo. The signal was filtered using a secondary low-pass Butterworth filter with a cutoff frequency of 20 Hz. Data from the force platform were recorded and analyzed (Kyowa sensor interface PCD-320A) to calculate the following variables: jump height (cm) = ((0.5 × flight time)2 × 2−1) × 9.81; maximum power (W) = (body weight + Fmax) × 9.81 × (flight time). The best performance in the jump height was used for further analysis. The ICC values for the jump height and power were 0.87 and 0.91, respectively.

#### 2.3.7. Wingate Test (Modified)

The peak power (PP) during the initial 10 s of the Wingate anaerobic test was measured on a mechanically braked bicycle ergometer (Monark Ergomedic 834E, Monark Vansbro, Sweden), 5–10 min after the CMJ testing. The external load was set at 0.075 kg·kg^−1^. After a 2 min warm-up on the bicycle ergometer, the athletes were instructed to pedal at 60 revolutions per minute for 2 min with light external resistance. During this time, two 3 s familiarization attempts were performed with the testing external load. Two minutes after this familiarization, the external testing resistance was applied, and the athletes continued to pedal at maximum voluntary speed for 10 s (the test was terminated 10 s after initiation) [15]. The athletes were verbally encouraged to pedal as fast as possible throughout the 10 s test duration. The number of revolutions was recorded in real time (1 kHz). Peak power was achieved 3–6 s after the application of the external resistance. The ICC for the Wingate PP in our laboratory was 0.92.

### 2.4. Statistical Analysis

Descriptive statistics were used for the statistical analysis: mean ± standard deviation. The normality of the data was assessed with the Shapiro–Wilk test, and no violations in normality were observed. The differences before and after the training period were analyzed with Student’s paired-sample *t*-Test. The Cohen’s d effect sizes were calculated. The correlation between the variables was examined with Pearson’s correlation coefficient. The level of significance was set at *p* ≤ 0.05. The reliability of all the measurements was tested using a two-way random effect intra-class correlation coefficient (ICC). The statistical analysis was performed with JASP software v. 0.18 (University of Amsterdam, The Netherlands).

## 3. Results

### 3.1. Differences between Pre and Post Training

The squat performance increased by 5.8 ± 7.0% (*p* < 0.05), bench press increased by 4.9 ± 9.8% (*p* = 0.05), deadlift increased by 8.3 ± 16.7% (*p* < 0.05), and total powerlifting performance increased by 6.5 ± 10.4% (*p* < 0.05, Table 1). The handgrip strength and CMJ performance/power were not altered by the training (Table 1). The peak power in the Wingate test was significantly increased by 10.9 ± 9.0% (*p* < 0.05). The body mass was increased by 2.2 ± 3.7% (*p* < 0.05). The lean body mass of the trunk region increased by 3.1 ± 4.7% (*p* < 0.05), but it was not altered significantly in any other body area. The bone mineral density, bone mineral content, body fat mass, body fat %, as well as visceral fat were not altered after the training period (Table 1). There was no statistically significant change in the VL muscle architecture and thickness. The CSA of all the quadriceps heads was not altered significantly after the training (Table 1).

### 3.2. Correlations at Pre and Post Training

Before the initiation of the 12-week preparation period (T1), the powerlifting performance was significantly correlated with the body mass, with the lean body mass of all the body parts measured, with the CSA of all the heads of the quadriceps, as well as with the VL thickness and pennation angle (Table 2). After the training period (T2), the powerlifting performance was significantly correlated with the body mass, with the LBM of all the body parts, with the CSA of all the heads of the quadriceps, as well as with the VL thickness (Table 3).

### 3.3. Correlations between Changes at pre and Post Training

The percentage change in the SQ performance was correlated with the percentage change in the body mass (r = 0.838, *p* = 0.001), the percentage change in the LBM of the lower extremities (r = 0.807, *p* = 0.003), the percentage change in the LBM of the trunk (r = 0.666, *p* = 0.025), the percentage change in the total body LBM (r = 0.782, *p* = 0.004, Figure 1), and the percentage change in the CSA of the quadriceps (r = 0.675, *p* = 0.023). The percentage change in the BP performance was correlated with the percentage change in the body mass (r = 0.764, *p* = 0.006) and the percentage change in the total LBM (r = 0.633, *p* = 0.037). The percentage change in the DL performance was correlated with the percentage change in the body mass (r = 0.803, *p* = 0.003), the percentage change in the LBM of the lower extremities (r = 0.665, *p* = 0.026), the percentage change in the LBM of the trunk (r = 0.670, *p* = 0.024), and the percentage change in the total LBM (r = 0.698, *p* = 0.017). The percentage change in the sum of all three lifts was correlated with the percentage change in the body mass (r = 0.868, *p* = 0.000), the percentage change in the LBM of the lower extremities (r = 0.744, *p* = 0.009), the percentage change in the LBM of the trunk (r = 0.713, *p* = 0.014), and the percentage change in the total LBM (r = 0.764, *p* = 0.006).

## 4. Discussion

The aim of this study was to investigate the relationship between the training-induced changes in LBM and muscle architecture and the changes in performance in well-trained powerlifters preparing for a competition. The main finding of this investigation was the significant positive correlations between the percentage increase in LBM and the percentage increase in powerlifting performance. Performance in powerlifting is determined by the maximum load lifted by the athlete. Besides anthropometric factors and neural activation, the quantity of skeletal muscle is perhaps the most important biological factor contributing to muscle strength [16]. Several studies have presented the close correlation between measures of muscle mass and powerlifting performance [4,5,7,17], which was also confirmed in the present study. However, this correlation is influenced by the bodyweight category of the athletes participating in these studies. Namely, as the bodyweight category of the athletes increases, the muscle mass is expected to increase and the muscle strength is expected to increase almost proportionally. Thus, the correlation between powerlifting performance and LBM at a specific time point, including athletes with large body mass variability, mostly presents the well-known relationship between muscle mass and strength. The main question for the present study was whether individual changes in LBM would be correlated with individual changes in powerlifting performance after a training period. The current data seem to support such a premise. The increase in SQ performance was highly correlated with the increase in LBM of the lower extremities and the trunk, which was anticipated because of the involvement of these body parts in the SQ [18]. In contrast, there was no significant correlation between the increase in SQ performance and LBM of the upper extremities, as was anticipated, because of the relatively small involvement of the upper extremities in the SQ. We believe that this set of correlations reinforces the current results. Moreover, the significant correlation between the increase in SQ performance and the increase in quadriceps’ CSA is along the lines of the current data. This also suggests that the quadriceps’ CSA measured with ultrasonography may be useful in estimating changes in SQ performance in well-trained powerlifters. At this point, it should be stressed that neither the quadriceps’ CSA nor the LBM (except of the trunk) was significantly altered following the training. In fact, for 2–3 athletes, there was a slight decrease in these parameters, and these athletes experienced small decrements in strength. This suggests that measurements of LBM and muscle CSA, together with changes in performance, should be systematically evaluated on an individual basis in powerlifters. 

The increase in BP performance was correlated with the increased change in total LBM but not the lean mass of the arms. It seems that the musculature of the body and the trunk are more important for small training-induced changes in well-trained powerlifters compared to the musculature of the arms. The increase in deadlift performance was correlated with the increase in LBM of the lower extremities and the trunk, which is in accordance with previous studies regarding the musculature involved in this resistance exercise [19]. One interesting finding of the present study was that the body mass was increased after the training period by 2.2 ± 3.7%, and this increase was strongly correlated with changes in powerlifting performance. It seems that this increase in body mass was the result of an LBM increase, although the latter did not reach statistical significance. Yet, these results may suggest that in well-nourished powerlifters preparing for a competition, determining the increase in body mass may be an effortless method for predicting performance changes. However, coaches should be careful with such increases in body mass since they might affect the athlete’s bodyweight category. Future studies should address this issue.

Both before and after the 12-week preparation period, the SQ and DL performances were significantly correlated with the CSA of all the heads of the quadriceps, as well as with the VL thickness. Muscle thickness at various body sites is highly correlated with performance in powerlifters [4]. Here, we reported that besides muscle thickness, the whole quadriceps’ CSA was also closely correlated with powerlifting performance, both at the beginning of the preparation cycle and in the competition. These results suggest that ultrasonography may be a useful tool for identifying differences in the quadriceps’ muscle mass among well-trained powerlifters and therefore estimating performance, although larger-scale studies need to confirm these data. 

The bone mineral density of powerlifters is the highest reported in sports [6]. Here, we reported high BMD and BMC values (Table 1). This is probably due to the high loading of the human skeleton with chronic resistance training. Previous studies revealed statistically significant increases in the BMD and BMC with resistance training, e.g., with high-intensity isokinetic resistance training of 5 months’ duration in young individuals [20]. However, in the current study, the athletes did not experience any change in BMD and BMC after the training, probably because of the already high initial values. Similarly, the body fat (both percentage and mass) was not altered following the training period. The athletes were competing in specific bodyweight categories; therefore, they managed their diet and training, aiming to avoid increases in body fat deposits. The visceral fat was higher than 1 kg in two of the participants. Visceral fat was significantly correlated with fat mass (r = 0.68, *p* < 0.05), and this suggests that powerlifters with increased body fat deposits should aim to reduce their body fat to reduce health-related risks associated with visceral fat stores [17]. 

The muscle architecture is an important parameter for strength and power performance. Unfortunately, this analysis is laborious and has been realized in few human skeletal muscles, mostly the VL. Muscle thickness was correlated with powerlifting performance at T1 and T2, but the training-induced changes in muscle thickness were not correlated with performance changes. This suggests that the measurement of the CSA of the quadriceps is a more informative parameter when evaluating changes in muscle mass and performance compared to the VL muscle thickness at a single time point in powerlifters. The pennation angle was correlated with powerlifting performance at T1, but not at T2. This is an interesting finding, which is difficult to explain, especially in light of the lack of correlation between these parameters, as revealed in a previous study [4]. The vastus lateralis fascicle length was not correlated with powerlifting performance, as also previously reported [4]. Moreover, the fascicle length in this muscle was not altered by the training, which suggests limited adaptations in this parameter with advanced powerlifting training. 

There was no increase in handgrip strength after the training period. It seems that these athletes had already achieved a high level of handgrip strength and further increases with 12 weeks’ training were not realized. Additionally, these athletes did not perform any specific training to increase their handgrip strength during this period, which supports the lack of handgrip strength increase. Similarly, there was no increase in lower body explosive strength (CMJ) after the 12-week training period, which is also probably the result of the lack of specific jumping training for the athletes. In contrast, there was an increase in the Wingate peak power of the lower extremities following the training. This was probably the result of the increase in lower body muscle strength of the athletes, which is shown to result in higher peak power in the Wingate test [15]. Compared to the CMJ, pedaling during the Wingate test is a much slower and longer-duration movement; therefore, there is more time to apply the increased strength achieved due to 12 weeks of powerlifting training. 

The increase in performance in each of the lifts was lower than that presented in a previous study (~11%) [21]. This discrepancy might be related to the lower initial performance level of the participants in that previous study. However, the relatively small number of participants and the inclusion of only a small number of female powerlifters may be a limitation of the present study. Moreover, despite the high correlations between LBM and strength, other parameters also influence powerlifting performance, such as the neural activation and anthropometrics (e.g., bone lengths), which were not measured. Another limitation of the present study is the lack of control of the athletes’ diet, which might have influenced the changes in body composition. Further studies should address these important issues. This research may be expanded in the future in quasi-experimental and/or experimental studies.

## 5. Conclusions

In conclusion, the current data revealed a high correlation between individual changes in LBM and performance in well-trained powerlifters following a 12-week training period before a competitive event. The changes in LBM provide an estimate of the changes in muscle mass, which is one of the most important biological parameters for powerlifting performance. Furthermore, measurement of the quadriceps’ CSA in powerlifters may also provide valid estimates of the muscle mass and SQ performance changes. In addition, the 12-week training carried out by the athletes achieved what it intended in terms of some variables (lifting performance, PP Wingate, trunk LBM and body mass), while in others, the minimum threshold of the difference between the T1 and T2 states was not reached.

## Figures and Tables

**Figure 1 jfmk-09-00089-f001:**
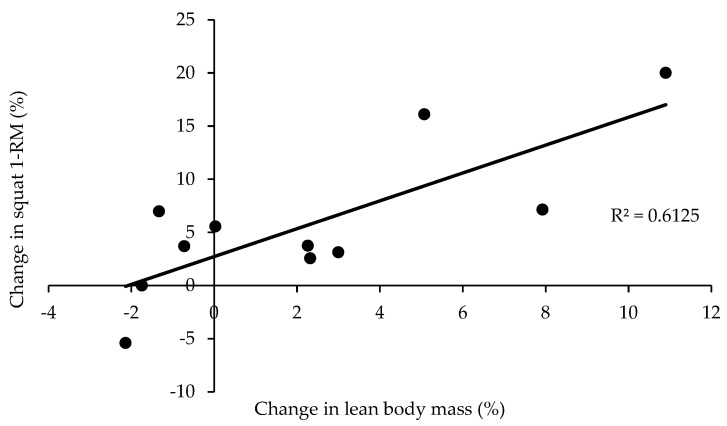
Correlation between the percentage change in the total lean body mass and the percentage change in the squat performance after 12 weeks of preparation for the national competition event in 11 powerlifters.

**Table 1 jfmk-09-00089-t001:** Performance, body composition and muscle architecture in 11 experienced powerlifters before and after 12 weeks of training in preparation for the national competition.

	T1	T2	Effect Size *d*	*p*
**Performance**				
SQ (kg)	198.3 ± 71.5	208.4 ± 73.1	1.203	0.003
BP(kg)	126.8 ± 44.7	132.7 ± 47.2	0.672	0.050
DL (kg)	221.4 ± 69.3	235.6 ± 68.6	0.749	0.030
Total lifts (kg)	546.6 ± 181.8	576.8 ± 184.4	0.992	0.008
Handgrip (sum, kg)	102.0 ± 30.7	102.5 ± 33.3	0.059	0.840
CMJ power (W·kg^−1^)	39.5 ± 3.4	39.5 ± 3.2	0.028	0.928
PP Wingate (W)	910 ± 255	1011 ± 303	1.236	0.002
**Body composition**				
Body mass (kg)	90.3 ± 21.6	92.2 ± 22.2	0.729	0.036
Fat mass (kg)	20.5 ± 10.1	21.1 ± 11.1	0.195	0.530
Fat (%)	23.3 ± 9.4	23.3 ± 9.6	0.017	0.950
LBM total (kg)	66.5 ± 16.6	67.9 ± 16.7	0.573	0.087
LBM arms (kg)	9.5 ± 3.3	9.7 ± 3.2	0.449	0.168
LBM trunk (kg)	31.1 ± 7.4	32.0 ± 7.7	0.675	0.049
LBM legs (kg)	22.5 ± 5.8	22.6 ± 5.6	0.155	0.619
BMC (kg)	3.251 ± 0.771	3.261 ± 0.760	0.262	0.406
BMD (g·cm^2^)	1.453 ± 0.178	1.455 ± 0.168	0.084	0.787
Visceral fat (gr)	421 ± 600	416 ± 652	0.067	0.830
**Vastus lateralis architecture**				
Pennation angle (^o^)	20.9 ± 3.3	20.7 ± 4.4	0.053	0.865
Fascicle length (cm)	8.1 ± 0.9	8.2 ± 1.0	0.106	0.733
Thickness (cm)	2.76 ± 0.46	2.83 ± 0.56	0.585	0.081
**Quadriceps’ CSA**				
VL (cm^2^)	32.2 ± 10.7	32.9 ± 10.4	0.522	0.114
RF (cm^2^)	7.2 ± 2.1	7.5 ± 1.9	0.610	0.071
VI (cm^2^)	35.4 ± 11.3	35.4 ± 10.2	0.019	0.951
VM (cm^2^)	19.2 ± 5.3	19.3 ± 5.1	0.015	0.961
Total CSA (cm^2^)	94.2 ± 28.4	95.2 ± 26.0	0.233	0.457

SQ = squat, BP = bench press, DL = deadlift, LBM = lean body mass, BMC = bone mineral content, BMD = bone mineral density, CSA = cross-sectional area, VL = vastus lateralis, RF = rectus femoris, VI = vastus intermedius, VM = vastus medialis, PP = peak power, CMJ = countermovement jump.

**Table 2 jfmk-09-00089-t002:** Correlations between powerlifting performance and body composition, vastus lateralis muscle architecture and quadriceps’ CSA at the initiation (T1) of the 12-week training preparation.

	Squat	Bench Press	Deadlift	Sum of Lifts
**Body composition**				
Body mass (kg)	0.807 **	0.774 **	0.779 **	0.805 **
LBM total (kg)	0.899 ***	0.915 ***	0.901 ***	0.923 ***
LBM arms (kg)	0.889 ***	0.930 ***	0.881 ***	0.915 ***
LBM trunk (kg)	0.884 ***	0.910 ***	0.890 ***	0.911 ***
LBM legs (kg)	0.881 ***	0.869 ***	0.881 ***	0.897 ***
**Vastus lateralis architecture**				
Pennation angle (^o^)	0.653 *	0.644 *	0.642 *	0.660 *
Fascicle length (cm)	0.418	0.374	0.438	0.424
Thickness (cm)	0.817 **	0.813 **	0.841 **	0.843 **
**Quadriceps’ CSA**				
VL (cm^2^)	0.870 ***	0.871 ***	0.907 ***	0.903 ***
RF (cm^2^)	0.786 **	0.837 ***	0.876 ***	0.849 ***
VI (cm^2^)	0.876 ***	0.890 ***	0.860 ***	0.900 ***
VM (cm^2^)	0.898 ***	0.795 ***	0.849 ***	0.873 ***
Total CSA (cm^2^)	0.914 ***	0.896 ***	0.911 ***	0.928 ***

LBM = lean body mass, CSA = cross-sectional area, VL = vastus lateralis, RF = rectus femoris, VI = vastus intermedius, VM = vastus medialis, * *p* < 0.05, ** *p* < 0.01, *** *p* < 0.001.

**Table 3 jfmk-09-00089-t003:** Correlations between powerlifting performance and body composition, vastus lateralis muscle architecture and quadriceps’ CSA following (T2) the 12-week training preparation.

	Squat	Bench Press	Deadlift	Sum of Lifts
**Body composition**				
Body mass (kg)	0.799 **	0.755 **	0.736 **	0.784 **
LBM total (kg)	0.921 ***	0.879 ***	0.943 ***	0.941 ***
LBM arms (kg)	0.923 ***	0.905 ***	0.950 ***	0.951 ***
LBM trunk (kg)	0.927 ***	0.885 ***	0.953 ***	0.949 ***
LBM legs (kg)	0.873 ***	0.823 **	0.886 ***	0.886 ***
**Vastus lateralis architecture**				
Pennation angle (^o^)	0.191	0.236	0.217	0.217
Fascicle length (cm)	0.557	0.565	0.540	0.566
Thickness (cm)	0.775 *	0.770 **	0.731 *	0.777 **
**Quadriceps’ CSA**				
VL (cm^2^)	0.828 **	0.831 **	0.790 **	0.835 **
RF (cm^2^)	0.809 **	0.818 **	0.799 **	0.827 **
VI (cm^2^)	0.873 ***	0.865 **	0.840 **	0.880 ***
VM (cm^2^)	0.836 ***	0.673 *	0.797 **	0.800 ***
Total CSA (cm^2^)	0.899 ***	0.866 ***	0.862 ***	0.899 ***

LBM = lean body mass, CSA = cross-sectional area, VL = vastus lateralis, RF = rectus femoris, VI = vastus intermedius, VM = vastus medialis, * *p* < 0.05, ** *p* < 0.01, *** *p* < 0.001.

## Data Availability

Data are available from the authors upon reasonable request.

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
