# Peer review of "Lean Body Mass, Muscle Architecture and Powerlifting Performance during Preseason and in Competition"

_jfmk, 2024, doi:10.3390/jfmk9020089_

Round 1
Reviewer 1 Report
Comments and Suggestions for Authors
The article objective is: investigate the changes in LBM and performance in powerlifters preparing for a competition.
The authors are requested to solve the following items:
1) Subsection 2.1 is described as “Experimental Approach”. As there are no independent and control groups, the research cannot be classified as experimental. Modify and describe the type of actual research used based on the data obtained and the manipulation of the variables analyzed.
2) It is recommended to start the Materials and Methods section with subsection 2.2 (Participants).
3) In subsection 2.4, the normality test used is not specified (describe it briefly), to justify the use of parametric statisticians, such as those used in the research.
4) Within the research limitations, recommend that research be expanded in the future into quasi-experimental and/or experimental studies.
Comments on the Quality of English LanguageConsult the text with English language specialists
Author Response
Reviewer 1
The article objective is: investigate the changes in LBM and performance in powerlifters preparing for a competition.
Response:
We thank the reviewer for taking the time to review our manuscript. We appreciate the efforts in providing us with comments that will enhance the quality of the paper and the message we are trying to deliver to the readership. We hope our replies and amendments are of satisfaction.
The authors are requested to solve the following items:
1) Subsection 2.1 is described as “Experimental Approach”. As there are no independent and control groups, the research cannot be classified as experimental. Modify and describe the type of actual research used based on the data obtained and the manipulation of the variables analyzed.
Response:
We thank the reviewer for the comment. We changed the subsection title to “Study design”. We also moved the current paragraph as second in methods according to the following comment of the reviewer.
2) It is recommended to start the Materials and Methods section with subsection 2.2 (Participants).
Response:
We thank the reviewer for the suggestion. We moved the subsection “Participants” at the start of the Materials and Methods.
3) In subsection 2.4, the normality test used is not specified (describe it briefly), to justify the use of parametric statisticians, such as those used in the research.
Response:
We thank the reviewer for the comment. We added the normality test.
4) Within the research limitations, recommend that research be expanded in the future into quasi-experimental and/or experimental studies.
Response:
We thank the reviewer for the suggestion. We added the reviewer’s suggestion into the limitations paragraph.
Reviewer 2 Report
Comments and Suggestions for Authors
This peer review article is a technological measure for building a training process in powerlifting. The combination of all research methods reveals the essence of changes in anthropometric indicators and physical fitness of athletes.
This article contains a logically structured pedagogical experiment with a clear description of each stage. A wide range of phenomena have been studied. A fairly complete analysis of the results obtained was carried out, which allows us to give an idea of ​​current research in this direction.
In this transition, questions relate to the approach to describing the results obtained. The limitations of the present study, as presented by the authors, relate to the overall results obtained.
As for the content of the comments, they should be presented in the following video:
Clarify information according to the terms of the declaration statement ( https://www.wma.net/what-we-do/medical-ethics/declaration-of-helsinki/doh-oct2008/ )
The content of the tests requires additional explanation for analysis. For this purpose, the following tests were used: Wingate test, Counter-movement jump.
It is necessary to reconsider the use of mathematical statistics methods in this device. The authors should add information about whether the elections were normally distributed, which would allow them to consider using the T-test and Pearson's p-test.
Due to the large variability of data, the question arises about the legality of using only average standard deviation values ​​when describing them. A better approach was to use nonparametric statistics to describe methods. For example, table 1: Visceral fat (g) 421±600.
It is advisable to change the title of Table 2 to a generalized definition of the indicators being studied.
In the section “Research Results” from the data in Table. 1, 2 it is unclear which contingent is represented – men or women. If a group has not been analyzed for six months, questions arise about the validity of this analysis due to the relationship that leads to differences in rates between women and men.
The author should check the data given in lines 283 – 288, according to the recommendations with the table. 1, 2. Special data for the upper limbs.
In lines 179 (Stasinaki et al. 2019) this source is missing.
Author Response
Reviewer 2
This peer review article is a technological measure for building a training process in powerlifting. The combination of all research methods reveals the essence of changes in anthropometric indicators and physical fitness of athletes.
This article contains a logically structured pedagogical experiment with a clear description of each stage. A wide range of phenomena have been studied. A fairly complete analysis of the results obtained was carried out, which allows us to give an idea of ​​current research in this direction.
In this transition, questions relate to the approach to describing the results obtained. The limitations of the present study, as presented by the authors, relate to the overall results obtained.
As for the content of the comments, they should be presented in the following video:
Clarify information according to the terms of the declaration statement ( https://www.wma.net/what-we-do/medical-ethics/declaration-of-helsinki/doh-oct2008/ ) The content of the tests requires additional explanation for analysis. For this purpose, the following tests were used: Wingate test, Counter-movement jump.
Response:
We thank the reviewer for taking the time to review our manuscript. We appreciate the efforts in providing us with comments that will enhance the quality of the paper and the message we are trying to deliver to the readership. We hope our replies and amendments are of satisfaction.
It is necessary to reconsider the use of mathematical statistics methods in this device. The authors should add information about whether the elections were normally distributed, which would allow them to consider using the T-test and Pearson's p-test.
Response:
We thank the reviewer for the comment. Yes, all data were normally distributed according to the result of the Shapiro Wilk test of Normality. We added this information in the statistics section.
Due to the large variability of data, the question arises about the legality of using only average standard deviation values ​​when describing them. A better approach was to use nonparametric statistics to describe methods. For example, table 1: Visceral fat (g) 421±600.
Response:
We thank the reviewer for the intriguing comment. Indeed, in some variables, such as the visceral fat, the SDs are large. However, the participants in this study are experienced powerlifting athletes and these data may be used from coaches and athletes as references values. In addition, previous studies on powerlifters followed similar statistical analysis (https://doi.org/10.1519/jsc.0b013e318269f81e, https://doi.org/10.1123/ijspp.2019-0974).
It is advisable to change the title of Table 2 to a generalized definition of the indicators being studied.
Response:
We thank the reviewer for the suggestion. The title of Table was changed accordingly.
In the section “Research Results” from the data in Table. 1, 2 it is unclear which contingent is represented – men or women. If a group has not been analyzed for six months, questions arise about the validity of this analysis due to the relationship that leads to differences in rates between women and men.
Response:
We thank the reviewer for the comment. Results in Table 1 and 2 include both male and female athletes. Furthermore, our discussion is mainly based on percentage change correlations; consequently, the validity of the analysis due to a potential difference in the proportion between female and male athletes’ adaptations is not affecting the nature of the results. [It should be mentioned that no significant difference was found between male and female athletes regarding the training-induced adaptations in powerlifting performance (p=0.399), lean mass (p=0.289) and all other variables].
The author should check the data given in lines 283 – 288, according to the recommendations with the table. 1, 2. Special data for the upper limbs.
Response:
We thank the reviewer for the comment. In these lines we refer to the percentage change correlations between variables and not to the correlations achieved during pre and post measurements.
In lines 179 (Stasinaki et al. 2019) this source is missing.
Response:
We thank the reviewer for pointing to us the missing reference.
Reviewer 3 Report
Comments and Suggestions for Authors
Dear authors, I congratulate you on your work. I think that the manuscript is very well prepared, the abstract gathers the necessary information to understand the background of the problem, the results and the conclusions obtained, the introduction describes the current state of recent research on the subject and the objective of the work is clearly defined. The methodology followed is consistent with the current state of research and follows all the necessary steps in a research study. The participants are few in number and especially the low number of women is a limiting factor of the research (you indicate this in the discussion). As the sport separates the competitions between men and women, in my view, the results for men should be shown separately from those for women, unless you reason to the contrary. I think this would be more relevant. The statistical analysis is appropriate for this type of study and the results are clearly represented.
The conclusions are well related to the results, although it could also be indicated that the training carried out by the subjects achieves what it intends in some variables (indicate which ones) while in others the minimum threshold of difference between the T1 and T2 states is not reached.
Best of luck to the authors.
Author Response
Reviewer 3
Dear authors, I congratulate you on your work. I think that the manuscript is very well prepared, the abstract gathers the necessary information to understand the background of the problem, the results and the conclusions obtained, the introduction describes the current state of recent research on the subject and the objective of the work is clearly defined. The methodology followed is consistent with the current state of research and follows all the necessary steps in a research study.
Response:
We thank you the reviewer for taking the time to review our manuscript. We appreciate the efforts in providing us with comments that will enhance the quality of the paper and the message we are trying to deliver to the readership. We hope our replies and amendments are of satisfaction.
The participants are few in number and especially the low number of women is a limiting factor of the research (you indicate this in the discussion). As the sport separates the competitions between men and women, in my view, the results for men should be shown separately from those for women, unless you reason to the contrary. I think this would be more relevant. The statistical analysis is appropriate for this type of study and the results are clearly represented.
Response:
We thank the reviewer for the intriguing comment and suggestion. The results in Tables 2 and 3 include both male and female athletes. However, the main purpose of the study was to investigate the percentage change correlations between all variables and powerlifting performance. Furthermore, no significant difference was found between male and female athletes regarding the training-induced adaptations. For example, the difference for powerlifting performance was p=0.399 and for total lean body mass was p=0.289. The lack of difference in the resistance training-induced adaptations between genders is also supported by previous studies in power athletes which have shown no significant differences between male and female athletes following long-term training programs (DOI: 10.1519/JSC.0000000000001048; Journal of Strength and Conditioning Research 17(4):p 739-745, November 2003).
The conclusions are well related to the results, although it could also be indicated that the training carried out by the subjects achieves what it intends in some variables (indicate which ones) while in others the minimum threshold of difference between the T1 and T2 states is not reached.
Response:
We thank the reviewer for the comment. We added the reviewer’s comment as suggested.
Best of luck to the authors.
We thank the reviewer.